# Association between Iron Deficiency and Survival in Older Patients with Cancer

**DOI:** 10.3390/cancers15051533

**Published:** 2023-02-28

**Authors:** Julie Tisserand, Violaine Randrian, Marc Paccalin, Pierre-Jean Saulnier, Marine Arviset, Arthur Fourmy, Victor Arriudarré, Amélie Jamet, Yvan Moreno, Simon Valéro, Evelyne Liuu

**Affiliations:** 1Geriatrics Department, Poitiers University Hospital, 86000 Poitiers, France; 2ProDiCeT, CHU Poitiers, Université de Poitiers, 86000 Poitiers, France; 3Gastroenterology and Hepatology Department, Poitiers University Hospital, 86000 Poitiers, France; 4University Hospital Poitiers, University of Poitiers, INSERM CIC 1402, 2 rue de la Milétrie, CEDEX, 86021 Poitiers, France

**Keywords:** geriatric oncology, iron deficiency, geriatric medicine, mortality

## Abstract

**Simple Summary:**

Iron deficiency is common in individuals with cancer. The aim of our study was to determine whether iron deficiency is associated with survival in older patients with solid tumors. We found that iron deficiency in the absence of anemia is associated with better survival. However, when combined with anemia, iron deficiency is associated with increased mortality. This study questions the value of iron supplementation therapy in older patients with cancer.

**Abstract:**

Background: iron deficiency (ID) is frequent in older patients. Purpose: to evaluate the association between ID and survival in patients ≥ 75 years old with confirmed solid tumors. Methods: a retrospective monocentric study including patients between 2009 and 2018. ID, absolute ID (AID) and functional ID (FID) were defined according to the European Society for Medical Oncology (ESMO) criteria. Severe ID was defined by a ferritin level < 30 µg/L. Results: in total, 556 patients were included, the mean age was 82 (±4.6) years, 56% were male, the most frequent cancer was colon cancer (19%, n = 104), and metastatic cancers were found in 38% (n = 211). Median follow-up time: 484 [190–1377] days. In anemic patients, ID and FID were independently associated with an increased risk of mortality (respectively, HR 1.51; *p* = 0.0065 and HR 1.73; *p* = 0.0007). In non-anemic patients, FID was independently associated with better survival (HR 0.65; *p* = 0.0495). Conclusion: in our study, ID was significantly associated with survival, and with better survival for patients without anemia. These results suggest that attention should be paid to the iron status in older patients with tumors and raise questions about the prognostic value of iron supplementation for iron-deficient patients without anemia.

## 1. Introduction

According to the World Health Organization (WHO), iron deficiency (ID) is the most common nutritional deficiency in the world. The consequences of ID in older patients are multiple, including anemia (in the United States, one out of five people aged 85 or over has anemia due to ID [1]), asthenia [2], immunodeficiency [3], and a reduction in cognitive functions [4]. The diagnosis of ID can be difficult to establish as it is frequently associated with inflammatory conditions, such as cancer, which interfere with the iron status. Two types of ID have been described: absolute iron deficiency (AID), which is a lack of iron reserves, and functional iron deficiency (FID), which reflects the unavailability of iron.

The effects of ID are particularly studied in cardiology settings. In chronic heart failure patients with reduced left ventricular ejection fraction (LVEF) (i.e., <50%), ID is associated with a significant increase in mortality, independently of anemia [5]. In this population, intravenous iron supplementation has yielded a reduction in hospitalizations for heart failure and improvements in exercise tolerance and quality of life [6], but these interventional studies did not find a reduction in mortality. Since 2016, martial supplementation has been recommended by the European Society of Cardiology (ESC) in heart failure patients with reduced LVEF. The 2021 recommendations go further [6], recommending acute I.V. iron supplementation during hospitalization for acute heart failure [7,8].

There are few data assessing the frequency of ID in the geriatric population, defined as aged 75 years or older.

A recent large-scale European study with more than 12,000 participants from three cohorts in different countries with a median age of 59 years found AID and FID, respectively, in 60% and 64% of participants. FID and severe AID (ferritin < 30 µg/L) were significantly associated with all-cause mortality [9,10,11,12].

Given that ID is a nutritional deficiency, and that malnutrition is a geriatric syndrome [13], ID may be more frequent in the older population with cancer [9]. Studies have addressed ID prevalence in patients with cancer [14,15,16]. In the CARENFER ONCO study, AID was reported in 20% and FID in 50% of patients with cancer [17]. A recent study estimates that 64% of patients diagnosed with colorectal cancer have ID [18]. However, few studies focus on older individuals. The aim of this study was to assess the relationship between ID, AID, FID, and survival in patients ≥ 75 years old with tumors.

## 2. Methods

### 2.1. Study Design and Population

This was a retrospective, observational, single-center study that included patients ≥75 years old, evaluated through a first geriatric oncology assessment at the University Hospital of Poitiers, France, between 1 January 2009 and 31 December 2018.

### 2.2. Data Collection

The data were collected prospectively by the geriatric oncology physician during the assessment. A second-stage data collection of biological variables was completed retrospectively.

The comprehensive geriatric assessment [10] included several validated tests to evaluate geriatric domains [19,20]: Activities of Daily Living (ADL), Instrumental Activities of Daily Living (IADL), Cumulative Illness Rating Scale (CIRS G), Mini Nutritional Assessment (MNA), Geriatric Depression Scale (GDS) with 15 questions, and Mini-Mental State Examination (MMSE). To assess the risk of falls, the following tests were administered: timed up and go, chair stand test, and the single-leg stance test.

We have grouped cancers at risk of bleeding, which included urological, gynecological, and digestive cancers [21], and excluded breast and prostate cancers [22].

We retrieved three comorbidities from medical records [23]: anemia according to WHO criteria (i.e., hemoglobin concentration less than 12 g/dL in women and less than 13 g/dL in men), chronic heart failure, and chronic kidney disease, defined by Cockcroft and Gault clearance < 60 mL/minute.

### 2.3. Outcomes

The primary outcome was to assess the relationship between ID, AID, FID, and survival in our population. We defined ID according to the 2018 European Society for Medical Oncology (ESMO) recommendations [24]: ID is defined by a TSAT (transferrin saturation) < 20%, AID by a ferritin level < 100 µg/L and FID by the association of a ferritin level ≥ 100 µg/L, and a TSAT < 20%. Severe ID [25] is defined by a blood ferritin value of less than 30 µg/L [26].

The duration of follow-up was defined from the day of the consultation with the geriatric oncology physician to the date of last news. The cut-off date was 22 January 2022.

This study was observational and was approved by the local ethics committee (2211250v0, reference: 23 January 2019), using existing data collected as part of the ANCRAGE cohort (Cancer and Age analysis).

### 2.4. Statistical Analysis

Quantitative variables are described by means, standard deviations, minimums, and maximums for variables with a normal distribution, and by medians and interquartiles for non-normal distribution variables. Categorical variables are expressed as numbers and percentages. Survival analyses were performed by the Log-rank test with a graphical representation according to the Kaplan–Meier method, and by the Cox proportional hazards model test. We used a Chi2 test to compare qualitative variables. The comparison of survival events was expressed as the hazard ratio (HR) and 95% confidence interval (95% CI). Variables significantly associated (*p* < 0.05) in univariate analysis were retained for multivariate analysis, with a stepwise regression. We carried out a descriptive and then a statistical analysis using Statview software (SAS Institute, version 5.0).

## 3. Results

### 3.1. Patients

During the study period, 556 patients were included in the study (Figure 1). Patients were predominantly male (n = 331, 56%), and the mean age was 82 years (Table 1). The prevalence of ID was 65%. ID was not associated with anemia in 36% of the cases. During the follow-up, 26% (n = 146) had at least one transfusion and 7% (n = 36) had at least one IV iron supplementation. Among these 36 patients, 4 had no anemia at the time of iron treatment.

Lack of iron assessment was the main reason for non-inclusion; the median age of these patients (n = 792) was 81 (78–85) years and 48% were female (n = 350). The most common cancer was breast cancer: 19% (n = 142). In 672 of those patients with available data, 40% (n = 228) had anemia.

#### 3.1.1. Iron Status

In our population, the prevalence of ID was high (65%); 25% (n = 136) had ID without anemia. The most frequent type of ID was FID. The prevalence of FID was 16% (n = 86) for patients without anemia and 27% (n = 147) for patients with anemia. The prevalence of AID was 11% (n = 62) in the absence of anemia and 16% (n = 89) with anemia. The prevalence of severe ID was 7% (n = 41).

#### 3.1.2. Oncological Characteristics

The main tumor sites were the colon and rectum (19% (n = 104)), prostate (14% (n = 78)) and skin (12% (n = 64)). Cancer was metastatic at the time of the geriatric assessment in 38% (n = 211) of patients. Advanced stage cancer was not more frequent in patients with iron deficiency compared to those with normal iron status (n = 139 patients with metastatic cancer and ID (38,8%) vs. n = 72 patients with metastatic cancer with normal iron status (38,5%), *p* = 0.94). The most common treatment was chemotherapy in 45% (n = 249) of cases.

### 3.2. Survival Analysis

#### 3.2.1. Survival According to General Characteristics

Univariate analysis

In univariate analysis (Table 2), survival was significantly associated with age (*p* = 0.036), anemia (*p* < 0.0001), chronic heart failure (*p =* 0.0314), chronic renal failure (*p =* 0.0035), cancers at risk of bleeding (*p* = 0.0150) and metastatic disease (*p* < 0.0001).

Concerning geriatric domains, autonomy (ADL, *p* < 0.0001 and IADL, *p* < 0.0001), comorbidities (CIRS-G, *p* = 0.0154), nutritional status (MNA, *p* < 0.0001), mood status (GDS-15, *p =* 0.0318), cognition (MMSE, *p =* 0.0349) gait and balance disorders “timed up and go” (*p =* 0.0009), chair stand test (*p =* 0.0002), and single-leg stance test (*p* = 0.0075) were all significantly associated with survival.

Multivariate analysis

In multivariate analysis (Table 2), age (*p* = 0.0404), nutritional status (MNA, *p* < 0.0001), anemia (*p* = 0.0209), and metastases (*p* < 0.0001) remained significantly associated with survival.

#### 3.2.2. Survival According to Iron Status

Univariate analysis

In univariate analysis, there was a significant association between survival and ID (*p* = 0.0106). Severe ID was significantly associated with survival (*p* = 0.0305).

In the absence of anemia, ID, AID and FID were associated with better survival (respectively, ID (HR 0.70, *p* = 0.0024), AID (HR 0.72, *p* = 0.0393) and FID (HR 0.73, *p* = 0.0244)) (Table 3).

Ferritin ≥ 100 µg/L and TSAT < 20% were significantly associated with survival (Figure 2).

Multivariate analysis

In anemic patients, ID (HR 1.51, *p* = 0.0065) and FID (HR 1.73, *p* = 0.0007) were associated with mortality. In non-anemic patients, only FID (HR 0.65; *p* = 0.0495) was associated with better survival. The nutritional status (MNA) and presence of metastases remained strongly associated with survival for both anemic and non-anemic patients (Table 4).

## 4. Discussion

In our study, which included 556 older patients with cancer, the prevalence of ID was high (65%, classified as severe in 7%). In more than one third of patients, ID was not associated with anemia.

Univariate and multivariate analyses have established a significant association between ID and survival, with better survival in non-anemic ID and increased mortality in subjects with both anemia and ID.

The most frequent types of primary cancer in our population were colorectal, prostate and skin, which differ from French national statistics compiled in 2017 [27]. The low number of patients with breast cancer in our study may be due to fewer referrals for this pathology by local gynecological surgeons and oncologists to the geriatrician. Women were likewise underrepresented in our population (44%).

Cancers identified as being at risk of bleeding (i.e., urogynecological and digestive cancers, excluding breast and prostate cancers) were frequent (52%), but no association was found with mortality. As risk of bleeding may be linked to stage progression, it would have been interesting to analyze this data in our study.

Nutritional status, assessed by the MNA score, was associated with better survival (HR 0.92 for each MNA point, *p* ≤ 0.0001), which is consistent with the literature [19,20].

A significant association was found between chronic heart failure and mortality in our population. This result corroborates those of a recent study evaluating the risk of developing cancer at 10 years using statistical models in patients with heart failure [28]. Data from the literature underline the frequent coexistence of cancer and heart failure, two conditions that have a major impact on life expectancy for older patients.

Anemia is a known predictor of mortality in cancer patients [29,30] and was significantly associated with mortality (HR 1.48; *p* = 0.0082), independently of oncological characteristics (type of primary cancer, cancers at risk of bleeding, and presence of metastasis).

Since non-anemic ID is an independent mortality factor in chronic heart failure patients with altered LVEF, we wondered whether this was validated in patients with cancer. Contrary to our initial hypothesis, ID was not systematically associated with an increased rate of mortality. Our study demonstrates the beneficial effect of ID in older patients with cancer without anemia.

Cellular and animal models have shown the effects of iron on tumor proliferation, through several signaling pathways and metabolic regulations [31]. Excess iron may be related to an increased risk of solid cancers such as colon, liver, stomach, kidney and lung cancers [32].

A pathophysiological explanation can be advanced, including several pathways.

Cancer cells use iron as a growth factor [33]. Tumor cells have a dysregulated iron metabolism and overexpress the transferrin receptor on their surface to increase the cellular uptake of iron. The tumor suppressor gene p53 induces cell cycle arrest, by decreasing intracellular free iron and inducing iron storage, via decreased expression of transferrin receptors to the cell membrane [32].

Iron also contributes to the maintenance of cancer stem cells [34]. Increased iron dependence has been reported in cancer cells and cancer stem cells in breast, ovarian, and prostate cancer cell models [35].

Iron is also involved in several epigenetic [36] and immunity [3] processes related to tumor initiation and progression.

The induction of ferroptosis could be an oncological therapeutic lead [37,38].

Intracellular free iron leads to the synthesis of reactive oxygen species (ROS) [39] via the Fenton reaction [29]. It produces oxidative compounds, which can damage DNA, and therefore contributes to the instability of the genome [40].

Hepcidin is a peptide synthesized by the liver. It regulates the concentration of iron in the blood. Its level is also dysregulated in cancer. Its secretion is stimulated in cases of inflammation, via interleukin 6 (IL6). It leads to the sequestration of iron in macrophages and enterocytes, resulting in a decreased level of serum iron. A study on breast cancer found that women with aggressive or chemo-resistant tumor phenotypes had higher levels of hepcidin [26]. Cancer cells also have the ability to synthesize their own hepcidin [41]. High hepcidin levels also lead to increased iron in cancer cells and activate Wnt and NF-κB signaling pathways, which are known to be linked to tumor progression [42,43].

Iron chelators have an antiproliferative effect via the inhibition of ribonucleotide reductase. This results in cell cycle arrest but also inhibits several cell signaling pathways involved in tumor progression [44]. Multiple in vitro studies evaluating the effects of iron deprivation observed cell cycle inhibition and apoptosis in colon, liver, and breast cancer cells [34]. Two iron chelators DFO (deferoxamine) and DFX (deferasirox) have demonstrated their efficacy alone or concomitantly with adjuvant chemotherapy in mice with gastric [45], esogastric, pancreatic, liver, and mammary cancer xenografts. Several clinical trials are underway to validate the therapeutic effect of iron chelators, particularly in breast cancer [46]. Other therapeutic strategies targeting iron metabolism are being evaluated, including monoclonal antibodies [47].

To our knowledge, this is the first study to assess the relationship between ID, AID, FID and survival in older patients with cancer. Studies have demonstrated that ID, with or without anemia, was associated with a worse prognosis and mortality, but did not take into account the type of ID [44]. A recent study demonstrated that FID seems to be independently associated with lymphatic invasion in colon cancers, with authors suggesting a potential relationship between FID and aggressive tumor characteristics [48]. The mean age of our population was 82 years, representative of the French sample of older patients with cancer. Furthermore, our study was based on a prospective cohort with consecutive patient recruitment. The retrospective collection of iron biological data was carried out blinded to the geriatric assessment and vital status. Our study is also valuable with prolonged patient follow-up (median duration of follow-up 484 days) and a large proportion of advanced disease, as more than one third of our population had a metastatic disease at the time of geriatric assessment.

This study has several biases. In terms of design, it is a retrospective and single-center study. There may be selection bias as patients referred to the geriatric oncology physician were previously screened by the oncologist, surgeon, or specialist according to a screening tool for geriatric frailty, as recommended by scientific societies, which excludes patients judged fit enough to undergo a typical cancer treatment plan [40,49]. Referral by the specialist was not systematic and was based on local habits. Thereby, our results cannot be extrapolated to the overall population of aged patients with cancer.

The results relied on a single biological assessment. Major biological variability is known for two biological parameters used to characterize ID, ferritin and TSAT. TSAT undergoes significant nychthemeral fluctuations ranging from 17 to 70% [50,51], reaching a maximum level in the morning. The level of ferritin may vary with inflammatory events. Iron status may also be closely correlated with bleeding events. Therefore, ferritin and TSAT could be poor markers for ID [52]. In addition, many patients were excluded due to the lack of biological data available for the iron assessment. These individuals were more frequently diagnosed with breast cancer and presented a lower rate of anemia. We wonder whether those subjects who did not have an iron status evaluation had a geriatric profile comparable to our population. For these patients, physicians may have limited the biological tests and focused on palliative care.

Given the current state of knowledge on the role of iron in oncology, it would be interesting to carry out a large prospective multicentric interventional randomized clinical trial to confirm those results and determine whether there is an interest in proposing intravenous iron supplementation in an iron-deficient patient with anemia or iron deficiency in the absence of anemia [36].

## 5. Conclusions

We identified a significant association between ID and survival in older patients with cancer. The retrospective nature of our study did not allow us to establish the main judgement criteria, which are, nevertheless, essential in geriatric oncology, i.e., the evaluation of the quality of life and resistance to effort. These results need confirmation in a large-scale prospective multicenter study that includes a large number of patients with various tumor sites and diverse geriatric profiles.

## Figures and Tables

**Figure 1 cancers-15-01533-f001:**
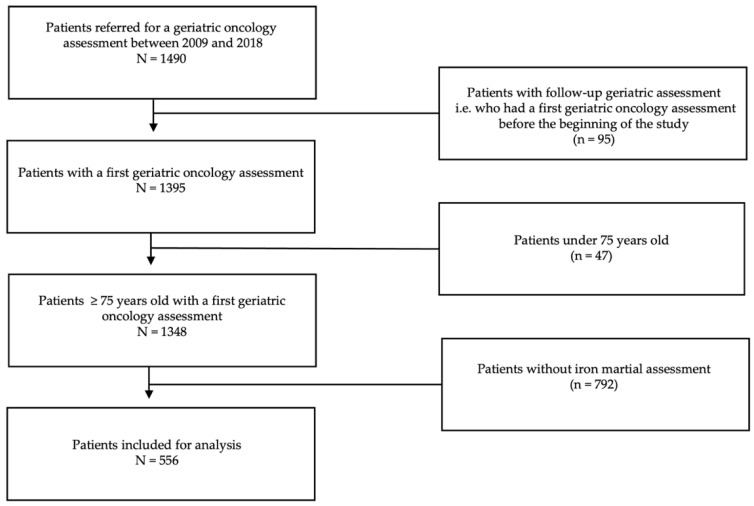
Flow chart.

**Figure 2 cancers-15-01533-f002:**
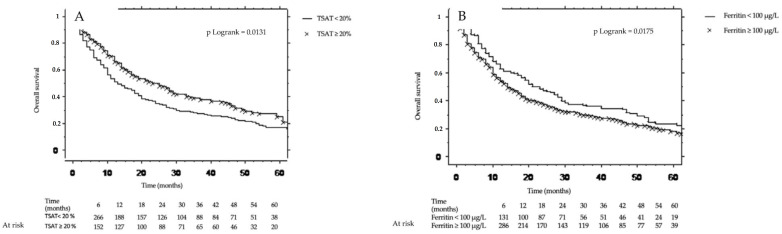
Kaplan–Meier curves according to transferrin saturation < 20% (**A**) and ferritin < 100 µg/L (**B**).

**Table 1 cancers-15-01533-t001:** Patient characteristics at baseline.

Variables	N, (%)
Gender, n = 556 (female)	245 (44)
Age, n = 556 (years), mean (S.D)	82 (±4.6)
Duration follow-up, n = 556, days (median) [interquartiles]	484 [190–1377]
*Geriatric assessment*	
ADL, n = 555, mean (S.D)	5 (±1)
IADL, n = 342, mean (S.D)	5 (±3.5)
Place of living, n = 539	
Community	472 (88)
Nursing home	40 (7)
Residential care home	10 (2)
Service home	2 (0.4)
Long-term care unit	1 (0.2)
CIRS-G, n = 556, mean (S.D)	8 (±5)
MNA, n = 553, mean (S.D)	21 (±6)
GDS 15, n = 425, mean (S.D)	4 (±3)
MMSE, n = 536, mean (S.D)	26 (±5)
Timed up and go, n = 533 (pathological)	151 (28)
Chair stand test, n = 459 (pathological)	170 (37)
Single-leg stance test, n = 492 (pathological)	256 (52)
Comorbidities	
Chronic heart failure, n = 556	159 (29)
Chronic renal disease, n = 528	323 (61)
Anemia, n = 545	302 (55)
*Iron measurements*	
Ferritin (µg/L), n = 545, mean (S.D)	381 (±366)
TSAT, n = 545	18 (±13)

ADL = Activities of Daily Living ADL, IADL = Instrumental Activities of Daily Living, CIRS-G = Cumulative Illness Rating Scale, MNA = Mini-Nutritional Assessment, GDS = Geriatric Depression Scale, MMSE = Mini-Mental State Examination, TSAT = Transferrin saturation, S.D = Standard Deviation.

**Table 2 cancers-15-01533-t002:** General characteristics related to overall survival.

Variables	Univariate Analysis			Multivariate Analysis		
	HR	[95% CI]	*p* *	HR	[95% CI]	*p* *
Gender, female	0.83	[0.69–1.01]	0.0683	-	-	-
Age, years	1.02	[1.00–1.04]	0.0306	1.03	[1.00–1.06]	**0.0404**
ADL (+1 point)	0.80	[0.74–0.86]	<0.0001	0.89	[0.75–1.06]	0.1969
IADL (+1 point)	0.90	[0.86–0.95]	<0.0001	0.97	[0.90–1.05]	0.5211
CIRS-G (+1 point)	1.02	[1.00–1.05]	0.0154	1.01	[0.98–1.05]	0.2798
MNA (+1 point)	0.89	[0.88–0.91]	<0.0001	0.92	[0.89–0.96]	**<0.0001**
GDS 15 (+1 point)	1.04	[1.00–1.07]	0.0318	1.01	[0.97–1.07]	0.4479
MMSE (+1 point)	0.97	[0.95–0.99]	0.0349	1.02	[0.99–1.06]	0.1404
Chronic renal failure	1.35	[1.11–1.65]	0.0027	1.04	[0.76–1.42]	0.7936
Chronic heart failure	1.24	[1.02–1.52]	0.0314	1.35	[1.00–1.84]	**0.0496**
Timed up and Go, pathological	1.42	[1.15–1.75]	0.0009	1.05	[0.64–1.73]	0.8323
Chair stand test, pathological	1.50	[1.21–1.86]	0.0002	0.95	[0.58–1.53]	0.8385
Single-leg stance test, pathological	1.31	[1.07–1.60]	0.0075	1.23	[0.88–1.70]	0.2112
Cancers at risk of bleeding **	1.26	[1.04–1.52]	0.0150	1.28	[0.95–1.72]	0.1038
Metastases	1.86	[1.54–2.25]	<0.0001	2.22	[1.65–2.98]	**<0.0001**
Anemia	1.71	[1.41–2.08]	<0.0001	1.41	[1.05–1.90]	**0.0209**

*p* * value for Cox model; Bold = significant *p* value at the threshold of 5%. ADL = Activities of Daily Living ADL, IADL = Instrumental Activities of Daily Living, CIRS-G = Cumulative Illness Rating Scale, MNA = Mini Nutritional Assessment, GDS = Geriatric Depression Scale, MMSE = Mini-Mental State Examination, TSAT = Transferrin saturation, I.V = Intravenous. ** Urogynecological cancers (excluding breast and prostate cancer) and digestive cancers.

**Table 3 cancers-15-01533-t003:** Iron status in relation to overall survival in univariate analysis.

Univariate Analysis									
Variables	Independently of Anemia			Without Anemia			With Anemia		
	HR	[95% CI]	*p* *	HR	[95% CI]	*p* *	HR	[95% CI]	*p* *
Severe ID	0.66	[0.45–0.96]	**0.0305**	0.70	[0.40–1.22]	0.2107	0.65	[0.40–1.07]	0.938
ID	1.29	[1.06–1.58]	**0.0106**	0.70	[0.56–0.88]	**0.0024**	1.69	[1.40–2.05]	**<0.0001**
AID	0.77	[0.63–0.96]	**0.0200**	0.72	[0.53–0.98]	**0.0393**	0.87	[0.67–1.13]	0.3126
FID	1.44	[1.19–1.74]	**0.0001**	0.73	[0.56–0.96]	**0.0244**	2.07	[1.96–2.55]	**<0.0001**

*p* * value for Cox model; Bold = significant *p* value at the threshold of 5%. ID = Iron Deficiency, AID = Absolute Iron Deficiency, FID = Functional Iron Deficiency.

**Table 4 cancers-15-01533-t004:** Iron status in relation to overall survival in multivariate analysis.

Multivariate Analysis											
Variables	Without Anemia			With Anemia							
	HR	[95% CI]	*p* *		HR	[95% CI]	*p* *		HR	[95% CI]	*p* *
FID	**0.65**	**[0.43–0.99]**	**0.0495**	**ID**	**1.51**	**[1.12–2.05]**	**0.0065**	**FID**	**1.73**	**[1.26–2.38]**	**0.0007**
Age	1.03	[0.99–1.06]	0.0588	Age	1.03	[0.57–1.51]	**0.0495**	Age	1.03	[0.99–1.06]	0.0759
ADL	0.89	[0.75–1.06]	0.2132	ADL	0.90	[0.76–1.07]	0.2714	ADL	0.92	[0.77–1.09]	0.3457
IADL	0.96	[0.89–1.04]	0.3834	IADL	0.96	[0.89–1.03]	0.3147	IADL	0.95	[0.88–1.03]	0.2533
CIRS-G	1.01	[0.98–1.05]	0.3033	CIRS-G	1.02	[0.98–1.05]	0.2311	CIRS-G	1.02	[0.99–1.06]	0.1281
MNA	0.92	[0.88–0.95]	**<0.0001**	MNA	0.93	[0.89–0.96]	**0.0004**	MNA	0.93	[0.89–0.97]	**0.0006**
GDS 15	1.02	[0.97–1.08]	0.3042	GDS 15	1.02	[0.97–1.07]	0.3756	GDS 15	1.02	[0.97–1.07]	0.3631
MMSE	1.03	[0.99–1.06]	0.0864	MMSE	1.02	[0.99–1.06]	0.1194	MMSE	1.02	[0.99–1.06]	0.1532
Timed up and go	1.07	[0.65–1.77]	0.7665	Timed Up and Go	1.06	[0.64–1.76]	0.8072	Timed Up and Go	1.01	[0.61–1.67]	0.9582
Chair stand test	0.93	[0.58–1.51]	0.7970	Chair stand test	0.93	[0.57–1.51]	0.7865	Chair stand test	1.00	[0.62–1.62]	0.9769
Single-leg stance test	1.23	[0.88–1.71]	0.2147	Single-leg stance test	1.22	[0.79–1.69]	0.2348	Single-leg stance test	1.21	[0.87–1.68]	0.2528
Chronic renal failure	1.13	[0.83–1.53]	0.4231	Chronic renal failure	1.08	[0.79–1.46]	0.6083	Chronic renal failure	1.08	[0.80–1.47]	0.5995
Chronic heart failure	1.44	[1.07–1.95]	**0.0162**	Chronic heart failure	1.33	[0.98–1.81]	0.0664	Chronic heart failure	1.41	[1.04–1.90]	**0.0243**
Cancers at risk of bleeding **	1.34	[0.99–1.80]	0.0515	Cancers at risk of bleeding **	1.21	[0.89–1.64]	0.2245	Cancers at risk of bleeding **	1.29	[0.96–1.74]	0.0877
Metastases	2.25	[1.67–3.02]	**<0.0001**	Metastases	2.16	[1.61–2.91]	**<0.0001**	Metastases	2.12	[1.58–2.85]	**<0.0001**

*p* * value for Cox model; Bold = significant *p* value at the threshold of 5%. ADL = Activities of Daily Living, IADL = instrumental Activities of Daily Living, CIRS-G = Cumulative Illness Rating Scale, MNA = Mini Nutritional Assessment, GDS = Geriatric Depression Scale, MMSE = Mini-Mental State Examination, TSAT = transferrin Saturation, FID = functional iron deficiency, ID = iron deficiency, ** urogynecological cancers (excluding breast and prostate cancer) and digestive cancers.

## Data Availability

The data presented in this study are available on request from the corresponding author.

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
