# Peer review of "Association between Iron Deficiency and Survival in Older Patients with Cancer"

_cancers, 2023, doi:10.3390/cancers15051533_

Round 1

Reviewer 1 Report

This was a retrospective, observational, single-center Study on the association between iron deficiency and

survival in 556 elderly patients with confirmed solid cancers.

The study is well-designed, the retrospective analysis seems robust, and the references are updated.

The pathophysiological mechanism should be further clarified.

Minor corrections required in the text:

Line 71, study instead of stud.

Line 274 to be corrected.

Lines 280-282 should be placed at the end of the conclusions underlying the limitations of this study.

Revision of language suggested.

Tables and charts seem redundant, authors should reduce their number.

In conclusion paper can be published with minor revisions.

Author Response

  • The pathophysiological mechanism should be further clarified.

We better clarified the pathophysiological mechanism:

  • (Line 270-272) “The tumor suppressor gene p53 induces cell cycle arrest, by decreasing intracellular free irons and inducing iron storage, via decreased expression of transferrin receptors to the cell membrane [36].”

  • (Line 286-293) “Iron chelators have an antiproliferative effect via inhibition of ribonucleotide reductase. This results in cell cycle arrest but also inhibits several cell signaling pathways involved in tumor progression [49]. Multiple in vitro studies evaluating the effects of iron deprivation observed cell cycle inhibition and apoptosis in colon, liver and breast cancer cells [44].Two iron chelators DFO (deferoxamine) and DFX (deferasirox) have demonstrated their efficacy alone or con-comitantly with adjuvant chemotherapy in mice with gastric [51], oesogastric, pancreatic, liver and mammary cancer xenografts. Several clinical trials are underway to validate the therapeutic effect of iron chelators, particularly in breast cancer [52]. Other therapeutic strategies targeting iron metabolism are being evaluated, including monoclonal anti-bodies [53].”

  • Minor corrections required in the text:

Line 71, study instead of stud. Line 274 to be corrected.

Lines 280-282 should be placed at the end of the conclusions underlying the limitations of this study.

Corrections were done as requested.

  • Revision of language suggested.

As suggested, we used the English editing service offering by MDPI.

  • Tables and charts seem redundant, authors should reduce their number.

We thank the reviewer for this comment. We have deleted three tables (Tables 1, 3 and 4)

Reviewer 2 Report

The manuscript reports the association between iron deficiency (ID) and mortality in older patients with cancer. This study is interesting, and the number of the patients seems to be large. However, my concerns are as follows.

#1: lines 27–28 and 196–198:

The authors report that "ID was significantly associated with mortality in older patients with cancer" and "cancers identified as being at risk of bleeding were frequent, but association with mortality was not found in multivariate analysis." However, I think the progression stage (early or advance) is more important to be analysed. Generally, advance stage of colorectal and upper digestive tract cancers tend to bleed more frequently than early ones. Therefore, I think most patients of ID group are advance stage (poor survival rate). Please clarify.

#2:

The lines 211–214 are hidden. Such error leads the study very unreliable.

#3: lines 54 and 132:

Please change "I.V" and "IV" to "intravenous".

Author Response

  • #1: lines 27–28 and 196–198:

The authors report that "ID was significantly associated with mortality in older patients with cancer" and "cancers identified as being at risk of bleeding were frequent, but association with mortality was not found in multivariate analysis." However, I think the progression stage (early or advance) is more important to be analysed. Generally, advance stage of colorectal and upper digestive tract cancers tend to bleed more frequently than early ones. Therefore, I think most patients of ID group are advance stage (poor survival rate). Please clarify.

Thank you for your relevant comment, suggesting that there could be a misunderstanding between bleeding risk cancers and stage progression. For clarification, we therefore decided to add the following sentence: “As risk of bleeding may be linked to stage progression, it would have been interesting to analyse this data in our study” (Line 249-250)

  • #2: The lines 211–214 are hidden. Such error leads the study very unreliable.

We are sorry for the layout problem. The layout shifted when we submitted the manuscript by the MDPI form.

  • #3: lines 54 and 132:

Please change "I.V" and "IV" to "intravenous

IV and I.V were changed to intravenous as requested.

Reviewer 3 Report

the Authors present the results of a well-conducted study evaluaing the association between ID and mortality in elderly patients with confirmed solid tumors. The Authors claim this is the first study to investigate this issue, but I urge the Authors to cross-check this assumption again - I found several studies that have addressed ID prevalence in several types of cancers. This should be made clear in the Introduction.

This said, the study can have relevance to clinical practice. I am only concerned by the way the Results are presented, almost with no text, only charts. can the Authors rework this section?

Last, the paper requires good editing in terms of English and style.

Author Response

  • The Authors claim this is the first study to investigate this issue, but I urge the Authors to cross-check this assumption again - I found several studies that have addressed ID prevalence in several types of cancers. This should be made clear in the Introduction.

We agree with this comment and added references that have addressed ID prevalence among cancer patients:

  1. Ploug, M.; Kroijer, R.; Qvist, N.; Lindahl, C.H.; Knudsen, T. Iron Deficiency in Colorectal Cancer Patients: A Cohort Study on Prevalence and Associations. Colorectal Disease 2021, 23, 853–859, doi:10.1111/codi.15467.
  2. de Castro, J.; Gascón, P.; Casas, A.; Muñoz-Langa, J.; Alberola, V.; Cucala, M.; Barón, F. Iron Deficiency in Patients with Solid Tumours: Prevalence and Management in Clinical Practice. Clin Transl Oncol 2014, 16, 823–828, doi:10.1007/s12094-013-1155-5.
  3. Martens, P.; Minten, L.; Dupont, M.; Mullens, W. Prevalence of Underlying Gastrointestinal Malignancies in Iron-Deficient Heart Failure. ESC Heart Failure 2019, 6, 37–44, doi:10.1002/ehf2.12379.
  4. Busti, F.; Marchi, G.; Ugolini, S.; Castagna, A.; Girelli, D. Anemia and Iron Deficiency in Cancer Patients: Role of Iron Replacement Therapy. Pharmaceuticals (Basel) 2018, 11, 94, doi:10.3390/ph11040094.
  • This said, the study can have relevance to clinical practice. I am only concerned by the way the Results are presented, almost with no text, only charts. can the Authors rework this section?

To improve the clarity of the presentation of our results, we reworked the section and removed three tables (Tables 1, 3 and 4). We also added the following text:

  • (Line 91-93) “ID is defined by a TSAT (Transferrin saturation) < 20 %, AID by a ferritin level < 100 µg/L and FID by the association of a ferritin level ≥ 100 µg/L and a TSAT < 20 %.”

  • (Line 162 -166) “In our population, the prevalence of ID was high (65 %), 25 % (n = 136) had ID without anemia. The most frequent type of ID was FID. Prevalence of FID was 16 % (n = 86) for patients without anemia and 27 % (n = 147) for patients with anemia. Prevalence of AID was 11 % (n = 62) in absence of anemia, and 16 % (n = 89) with anemia. Prevalence of severe ID was 7% (n = 41).”

  • (Line 169-171) “The main tumor sites were colon and rectum 19 % (n = 104), prostate 14 % (n = 78) and skin 12 % (n = 64). Cancer was metastatic at the time of the geriatric assessment in 38 % (n = 213) of patients. The most common treatment was chemotherapy in 45 % (n = 249) of cases.”
  • Last, the paper requires good editing in terms of English and style.

As suggested, we used the English editing service offering by MDPI.

Round 2

Reviewer 2 Report

Thanks for your revision. However, my main question (comment #1) is not resolved yet. I strongly think that the ID group have more patients of advanced stage cancer than the control group. Therefore, the authors must clarify the number of advance or early stages of those groups (all 556 patients) shown in Table.

Author Response

  • Thanks for your revision. However, my main question (comment #1) is not resolved yet. I strongly think that the ID group have more patients of advanced stage cancer than the control group. Therefore, the authors must clarify the number of advance or early stages of those groups (all 556 patients) shown in Table.

Thank you for this comment.

We used a Chi2 test and added the following sentences:

  • (Line 105-106) “We used Chi2 test to compare qualitative variables.”
  • (Line 171-174) “Cancer was metastatic at the time of the geriatric assessment in 38% (n = 211) of patients. Advanced stage cancer was not more frequent in patients with iron deficiency compared to those with normal iron status (n= 139 patients with metastatic cancer and ID (38,8%) vs n= 72 patients with metastatic cancer with normal iron status (38,5%), p = 0,94).”

Reviewer 3 Report

thanks for having considered my comments

Author Response

Thank you for your valuable comments and your validation of this revised manuscript

Round 3

Reviewer 2 Report

Thanks for your revision. It is su